# Design and Realization of Ultra-Wideband Differential Amplifiers for M-Sequence Radar Applications

**DOI:** 10.3390/s24072143

**Published:** 2024-03-27

**Authors:** Miroslav Sokol, Pavol Galajda, Patrik Jurik

**Affiliations:** Department of Electronics and Multimedia Telecommunications, Technical University of Košice, 042 00 Kosice, Slovakia; pavol.galajda@tuke.sk (P.G.); patrik.jurik@tuke.sk (P.J.)

**Keywords:** ultra-wideband, UWB, differential, amplifier, ASIC, SoC, radar

## Abstract

Amplification of wideband high-frequency and microwave signals is a fundamental element within every high-frequency circuit and device. Ultra-wideband (UWB) sensor applications use circuits designed for their specific application. The article presents the analysis, design, and implementation of ultra-wideband differential amplifiers for M-sequence-based UWB applications. The designed differential amplifiers are based on the Cherry–Hooper structure and are implemented in a low-cost 0.35 µm SiGe BiCMOS semiconductor process. The article presents an analysis and realization of several designs focused on different modifications of the Cherry–Hooper amplifier structure. The proposed amplifier modifications are focused on achieving the best result in one main parameter’s performance. Amplifier designs modified by capacitive peaking to achieve the largest bandwidth, amplifiers with the lowest possible noise figure, and designs focused on achieving the highest common mode rejection ratio (CMRR) are described. The layout of the differential amplifiers was created and the chip was manufactured and wire-bonded to the QFN package. For evaluation purposes, a high-frequency PCB board was designed. Schematic simulations, post-layout simulations, and measurements of the individual parameters of the designed amplifiers were performed. The designed and fabricated ultra-wideband differential amplifiers have the following parameters: a supply current of 100–160 mA at −3.3 V or 3.3 V, bandwidth from 6 to 12 GHz, gain (at 1 GHz) from 12 to 16 dB, noise figure from 7 to 13 dB, and a common mode rejection ratio of up to 70 dB.

## 1. Introduction

Differential wideband amplifiers and their modifications are indispensable in UWB sensor systems. “With the invention of television and radar during World War II, the design of broadband amplifiers proved to be very important. In today’s digital world, this is even more true. It is a paradox that designers of analog and digital devices are still dependent on oscilloscopes that, at least in their input section, consist of analogue broadband amplifiers [1]”. These wideband amplifiers are used as transmitting amplifiers to amplify the transmitted signal, receiving amplifiers for amplification of the received signal, or as special-purpose amplifiers. Using differential signaling has one general undeniable advantage—it is very effective in terms of removing common-mode noise or interference.

Common-mode noise is defined as the noise situated in both signals of a differential pair. If we assume that the differential signal pair is formed symmetrically and close to each other, we can say that the noise will be the same for both signals. The ideal assumption of the suppression of the common-mode signal (CMRR parameter) by the differential amplifier defines the elimination of signal interference as [2]:(1)$$V_{diff}=\left(V_{d+}+V_{noise}\right)-\left(V_{d-}+V_{noise}\right)=V_{d+}-V_{d-},$$
where $V_{diff}$ is the output voltage of the amplifier, $V_{d+}$ and $V_{d-}$ are positive and negative signals, respectively, and $V_{noise}$ is the noise applied to the differential pair. In practical terms, using a differential guided analog signal or digital communication provides larger resistance to interference. Signals that are differentially routed emit less signal radiation into the surrounding environment when compared to single-ended routing. This has a positive effect on reducing interference [3].

In the field of amplifiers, there are not many references relating to amplifiers designed directly for UWB sensor systems operating in wideband frequencies up to 13 GHz which are designed using low-cost 0.35 µm SiGe BiCMOS technology. There are many structures for telecommunication and optical networks [4,5,6,7,8,9,10,11,12], but most of the time these amplifiers are designed with expensive semiconductor technologies, and even though they are wideband, by definition, their bandwidth does not meet the requirements for UWB sensor systems based on the M-sequence. Applications of UWB sensor systems based on the M-sequence that have been developed include, for example, ground penetrating radars [13,14], locating and searching for persons behind obstacles and walls [15], locating general objects or use of robots [16], in the field of non-invasive diagnostics and condition detection in medicine [17], and material reflectometry [18,19]. New circuit structures have also been developed for UWB sensor systems, such as transceivers [20,21], wideband couplers [18], and AD converters [22]. With the development of the new circuits came the need to design additional differential amplifiers directly for these purposes and specific implementations. In the past, monolithic structures of differential amplifiers have been developed at our department [23]. Another reason for designing new amplifiers is to determine the improvement in parameters over the previous amplifiers. There is room for improvement also in the design of layouts for manufacturing, where we can significantly improve parameters by optimizing the layout design without changing the schematic structure. In addition, with the newer design tools, optimizing cell layouts is easier than in the past.

This paper summarizes the basic theoretical knowledge of the functioning of differential structures and their auxiliary circuits, such as current mirrors and input-output circuits. This theoretical knowledge is applied to the design of differential amplifiers and their auxiliary circuits. Further, various solutions and designs of differential amplifiers, input matching circuits, output circuits, and current mirrors, adjusting the operating points of the amplifiers, are presented. Last but not least, the solutions derived and the evolution of the layout of the proposed amplifiers are presented. At the end of the paper, the individual designs are evaluated and compared.

## 2. Use of a Cherry–Hooper Differential Amplifier

As already mentioned, there are many types of differential amplifiers. Not every type is suitable for a specific UWB sensor system application or a specific semiconductor manufacturing process. Each design represents a compromise regarding certain parameters. In a wideband amplifier, we can consider two interrelated opposing parameters, namely, the bandwidth and the gain of the amplifier. For example, in a simple differential amplifier, increasing the bandwidth leads to a decrease in the gain and the dynamic range. Similarly increasing the gain at a constant supply voltage leads to a decrease in the dynamic range. The Cherry–Hooper amplifier reduces these dependencies because it consists of two amplification stages. The basic structure was designed by E.M. Cherry and D.E. Hooper [24,25]. These amplifier stages create an impedance difference between stages [24]. The structure of the Cherry–Hooper amplifier is designed to ensure that the adjustable gain, the bandwidth, and the dynamic range of the amplifier are less interdependent.

Cherry–Hooper amplifiers are mostly used in applications in digital optical communications [26], but due to the larger bandwidth compared to a standard differential amplifier, differential amplifiers based on the Cherry–Hooper structure are also promising for use in UWB radar systems emitting the M-sequence. The Cherry–Hooper-type amplifiers described in this paper are primarily designed for special types of directional couplers [27,28].

In the original basic structure, simple passive resistive feedback was implemented [6] but this passive feedback had the disadvantage of loading the output and, thus, reducing the dynamic range of the amplifier. By changing the resistive feedback to feedback created by a unity-gain emitter follower, the impedance decouples the output without affecting the transmission of this feedback. The basic structure of a modified Cherry–Hooper differential amplifier with active feedback is shown in Figure 1. The replacement of passive feedback with active feedback improves the dynamic properties of the amplifier and reduces the trade-off between the gain settings and the frequency bandwidth, or the frequency bandwidth and the dynamic range. The resistors $R_{1}$ and $R_{2}$ form a voltage divider for the emitter follower input and, together with the resistor $R_{f}$, set the feedback. The emitter follower is formed by a transistor $Q_{5}$ ($Q_{6}$). This makes it possible to change the gain of the amplifier without changing the resistor $R_{f}$ or the current $I_{T1}$.

To assess the small-signal gain, the half-circuit equivalent of the proposed amplifier was analyzed (see Figure 2). By neglecting parasitic effects and simplifying bipolar transistor models, it is possible to determine the gain for small signals of the first $A_{1}$ and the second $A_{2}$ stage of the differential amplifier:(2)$$A_{1}=-\frac{v_{o1}}{v_{id}/2}=-\frac{g_{m1}\left(R_{f}+1/g_{m5}\right)}{1+R_{1}g_{m3}}$$
(3)$$A_{2}=-\frac{v_{od}/2}{v_{o1}}=-g_{m3}\left(R_{1}+R_{2}\right)$$
Equation (Equation 2) shows that by increasing the transconductance of the second stage $gm_{3}$ or the value of the resistor $R_{1}$, the gain in the first stage is reduced. The resulting gain of the whole differential stage for small signals is given by $A_{d}\;=\;A_{1}A_{2}$. By multiplying Equations (Equation 2) and (Equation 3), the resulting gain is given by:(4)$$A_{d}=\frac{v_{od}}{v_{id}}=\frac{g_{m1}\left(R_{1}+R_{2}\right)\left(1/g_{m5}+R_{f}\right)}{\left(1/g_{m3}+R_{1}\right)}$$
The gain of the amplifier can be increased by increasing the current $I_{T1}$ and $g_{m1}$, which also increases the output voltage. It is possible to adjust the gain without significantly affecting the maximum output voltage amplitude by changing the ratio of the resistors $R_{1}$ and $R_{2}$. Maximizing the amplifier’s bandwidth is possible by changing the ratio of the resistors $R_{f}/R_{1}$, without significantly affecting the gain and dynamic range. A more detailed analysis of the dependence of the individual parameters on the change in resistor values is presented in [6].

## 3. Selection of Current Sources

The development of differential amplifiers in integrated form also implies the development and use of suitable current sources. For all the proposed circuits in this paper, current sources with NMOS transistors have been used. When selecting current sources, larger variability in the length/width ratio setting W/L and lower saturation voltage of NMOS transistors are possible. The advantage of using these is that the NMOS current mirrors are voltage-regulated which eliminates the gain error of the current mirror, as opposed to the current source with current controlled by bipolar transistors [29].

In Figure 3a the simplest form of the current mirror with NMOS transistors is shown. The transistor $M_{1}$ has a gate connected to the drain and operates in a saturation or linear mode depending on the voltage $V_{GS1}$. If the transistor $M_{2}$ works in the active region, then by the voltage $V_{GS2}$, which is equal to the voltage $V_{GS1}$, it is possible to control the current $I_{D2}$ according to Equation [30]:(5)$$V_{GS2}=V_{t}+\sqrt{\frac{2I_{D2}}{k^{\prime }{\left(W/L\right)}_{2}}}=V_{GS1}=V_{t}+\sqrt{\frac{2I_{D1}}{k^{\prime }{\left(W/L\right)}_{1}}},$$
where $k^{\prime }$ is the value of transconductance of NMOS and is specified by the production process. The value of $k^{\prime }$ for NMOS transistors in 0.35 µm SiGe BiCMOS technologies is approximately *W* is the width, *L* is the transistor channel length, and $V_{t}$ represents the threshold voltage of the NMOS transistor. 140 µA/V up to 200 µA/V [29,30],

If the transistors are the same, i.e., ${\left(W/L\right)}_{2}={\left(W/L\right)}_{1}$, we can assume that:(6)$$I_{OUT}=I_{D2}=I_{D1}$$
For the same transistors operating in the active region, the current mirror has a unity gain. In practice, the transistor $M_{1}$ and the transistor $M_{2}$ may have different dimensions. Using Equations (Equation 5) and (Equation 6), we can determine the ratio of currents with respect to the dimensions of the NMOS transistors as follows:(7)$$I_{OUT}=\frac{{\left(W/L\right)}_{2}}{{\left(W/L\right)}_{1}}I_{D1}$$
Equation (Equation 7) shows that the gain of the current mirror can be greater or less than one because the size of the transistors can be varied. The minimum saturation output voltage $V_{sat}$ of the current mirror with NMOS transistors can be determined by Equation (Equation 8) [29]:(8)$$V_{sat}=V_{GS2}-V_{t}=\sqrt{\frac{2I_{OUT}}{k^{\prime }{\left(W/L\right)}_{2}}}$$

From Equation (Equation 8), it can be concluded that $V_{sat}$ depends on the geometry of the transistor and can be arbitrarily small. However, from the technological characteristics of the NMOS transistor, it can be noted that the minimum threshold voltage is defined as $V_{sat}=2nV_{T}$, where $V_{T}$ is the thermal voltage [29], and $n=(C_{js}/C_{ox})+1$. For 0.35 µm, SiGe BiCMOS technology has gate oxide capacitance $C_{ox}=\;4.6\;\mathrm{fF}/\mu m^{2}$ and junction capacitance $C_{js}=\;0.84\;\mathrm{fF}/\mu m^{2}$. Based on this, we can assume that n=1.18 and $V_{sat}=2nV_{T}=59\;\mathrm{mV}$, at room temperature 25 °C.

Nevertheless, there is more variability when adjusting the width of the NMOS transistor compared to the size of a bipolar transistor; in other words, with NMOS transistors at the same current source, we can achieve a lower minimum saturation voltage of the current mirror, which has a positive effect on setting the operating points of the designed circuits. Figure 4 shows the output characteristic of a current mirror with NMOS transistors. It can be seen that if the voltage $V_{OUT}$ drops below the minimum saturation voltage $V_{sat}$, the current source drops out of saturation mode and loses its functionality.

As part of the development, the reference resistor *R* (Figure 3a) was replaced by an NMOS transistor $M_{3}$ (see Figure 3b). This had the advantage of saving space when designing the layout of the individual amplifiers. This circuit creates a quadrature voltage divider from two NMOS transistors [31]. The transistor $M_{3}$ operates in saturation mode, which is determined by the condition $V_{DS3}<V_{GS3}-V_{t}$, where $V_{t}$ is the threshold voltage of the NMOS transistor. The currents $I_{D3}$ and $I_{D1}$ flow through transistors $M_{3}$ and $M_{1}$ in saturation mode and they are equal, given by the equations [29]:(9)$$I_{D3}=\frac{k^{\prime }}{2}{\left(\frac{W}{L}\right)}_{3}{\left(V_{GS3}-V_{t}\right)}^{2}=I_{D1}=\frac{k^{\prime }}{2}{\left(\frac{W}{L}\right)}_{1}{\left(V_{GS1}-V_{t}\right)}^{2}$$
Modifying Equation (Equation 9) and substituting the variables according to Figure 3b, we obtain: (10)$${\left(\frac{W}{L}\right)}_{3}{\left(V_{DD}-V_{IN}-V_{t}\right)}^{2}={\left(\frac{W}{L}\right)}_{1}{\left(V_{IN}-V_{t}\right)}^{2}$$
If we know the $V_{t}$ values of both transistors and the supply voltage $V_{DD}$, it is possible from the relations (Equation 9) and (Equation 10) to say that the reference current $I_{D3}$ and the voltage $V_{IN}$ can be adjusted using the width-to-length ratio of the transistors $M_{3}$ and $M_{1}$. From the above theoretical analysis of the NMOS current mirror and referring to [29,30], it is possible to say that current mirrors with NMOS transistors provide greater setup variability at small supply voltages and provide lower saturation voltage values compared to bipolar transistors. Another advantage of current mirrors built from NMOS transistors is the possibility of connection of multiple outputs without degrading the mirror gain, as is the case with bipolar mirrors; thus, it is possible to drive several outputs with many times larger transistors using one small NMOS. Based on this analysis, it can be said that current mirrors based on NMOS transistors are better suited for practical applications in BiCMOS technology.

## 4. Input and Output Matching Circuits

For high-frequency and wideband applications, the input and output matching of the circuits is very important. Matched inputs reduce losses and also reduce standing waves due to mismatch [32]. Emitter followers are extensively used as input and output circuits. The common identifying feature of this circuit is that the signal is applied to the base and the output is taken from the emitter. A basic schematic of the emitter follower is shown in Figure 5a. In terms of signal integrity, the output signal is identical to the input signal, but shifted by the DC value of the voltage *V_BE_*. Because the voltage *V_BE_* is a logarithmic function of the collector current *I_C_*, the base-emitter voltage *V_BE_* is almost constant, although the collector current varies over a larger range. An equivalent small-signals model is shown in Figure 5b. The emitter follower is characterized by a unit voltage gain, which can be based on the conditions $\beta_{0}\gg 1$ and $r_{0}\gg R_{L}$ [29] expressed as:(11)$$A_{v}=\frac{v_{o}}{v_{i}}⋍\frac{g_{m}R_{L}}{1+g_{m}R_{L}},$$
where *R_L_* is the value of the load resistance or the output resistance of the current source.

It is also possible to approximate the input resistance of the emitter follower $R_{i}$ by replacing the voltage source with a current source $i_{t}$. The resulting input resistance can be expressed as [29]:(12)$$R_{i}=\frac{v_{t}}{i_{t}}=r_{\pi }+\left(\beta_{0}+1\right)\left(R_{L}\|r_{0}\right)$$

Based on Equation (Equation 12), it can be said that the emitter follower is characterized by high input resistance, which is given by the base resistance $r_{\pi }$ and output resistance multiplied by the current gain $\beta_{0}$ of the transistor. The advantage of the high input impedance in order units of kΩ of the emitter follower is that it is possible to adjust the input impedance using resistors according to the requirements of the high-frequency circuits. In Figure 6a–c are shown the input circuits used in the design of the wideband amplifiers described in this paper. Based on the small-signal equivalent circuit, we can approximate the input wideband impedance of all three inputs. A small-signal equivalent schematic for wideband signals is shown in Figure 7.

Input impedance $Z_{in}$ of the emitter follower based on Equation (Equation 12) and $\beta_{0}=r_{\pi }g_{m}$ can be expressed as [29]: (13)$$Z_{in}=\left(r_{\pi }+\left(1+r_{\pi }g_{m}\right)\left(R_{L}\|r_{0}\right)\right)\|\frac{1}{j\omega C_{in}},$$
where $C_{in}$ represents the input capacity of the emitter follower:(14)$$\;C_{in}=C_{\mu }+\frac{C_{\pi }}{1+g_{m}R_{L}}$$
The resulting differential and single-ended input impedance values can be approximated as a parallel combination of the emitter follower input impedance and the impedance defined by the type of external circuit.

**Input matching type I** is used to set the input differential impedance regardless of the single-ended impedance. Resistors $R_{2}-R_{5}$, as shown in Figure 6a, have values of the order of kΩ so that they do not create additional power consumption and are intended to set the operating point of the next amplifier. Using resistors $R_{2}-R_{5}$, this operating point is set to the approximate center of the input range of the amplifier. To set, the operating point must be offset by *V_BE_*. The resistor, $R_{1}$
Figure 6a, adjusts the differential input impedance of the amplifier for a particular application. This input matching has been used for amplifiers designed for wideband directional couplers, where the input differential impedance was set to 50Ω. This type of input is suitable only if differential signaling is assumed. In this case the nominal value of the input impedance is set to 100Ω [3].

**The input matching type II** is the simplest in terms of circuitry, as shown in Figure 6b. It consists of two resistors $R_{1}$ and $R_{2}$, which are directly connected to the ground. In this case the input single-ended impedance for each of the inputs is set. The differential impedance is $R_{1}+R_{2}$. This type has the advantage that the single-ended impedance value for each input is set at the same time, as is the differential impedance between the inputs.

Another advantage is the wideband capability; in this case, we can start at the DC signal up to high frequencies. The disadvantage is the direct connection of $R_{1}$ and $R_{2}$ to the ground, which, in terms of operating point settings, may not suit the next amplifier settings. Sometimes it is necessary to connect up to two emitter followers in a row to achieve a shift in DC voltage from input to output by $2\;V_{BE}$. Another disadvantage is that it requires a symmetrical or negative supply voltage in order to create an operating point to operate the transistors in active mode. In the case of positive supply only, the input impedance is tied to the supply *VCC* and coupled to *GND* through the bypass capacitors in the circuit, which can cause different unwanted crosstalk and current loops in the circuit. The input matching type II was used in the input amplifier for the clock signal in the designed 7-bit ADC [22].

**Input matching type III** combines the advantages of type I and II. The circuit in Figure 6c maintains the operating point setting; i.e., using resistors $R_{3}-R_{6}$, it is possible to adjust the DC input voltage as required regardless of the type of power supply. The differential impedance is set by a series connection of resistors $R_{1}$ and $R_{2}$. Ideally, i.e., after neglecting the input impedance of the emitter follower and the parasitic properties of the chip, the differential impedance in such an arrangement is constant and independent of frequency. The single-ended impedance of each input coupled to the ground depends on the value of $R_{1}$ and $R_{2}$ and the capacitances of $C_{1}$ and $C_{2}$, respectively. In this case, the capacitances $C_{1}$ and $C_{2}$ represent a “short-circuit” for the high-frequency signal. The disadvantage of such wiring is that the non-differential matching of inputs depends on the size of the capacitances $C_{1}$ and $C_{2}$. Within the chip design, it is possible to use capacitances of sizes of the order of units of pF, allowing single-ended input matching at the start frequencies of the order of hundreds of MHz to units of GHz. Such wiring can be used for applications that require input signals higher than hundreds of MHz, for example, clock or mixer input. However, it is possible to lead a contact from the chip to connect the capacitances and supplement these capacities with an external off-chip capacity. This will improve matching to units of MHz but will increase the number of external components around the chip.

A comparison of input matching of different types of input circuits is shown in Figure 8b. The comparison represents a schematic simulation without the influence of the parasitic effects of the chip. Figure 8a shows the single-ended 50Ω input matching. It can be seen that in the case of the first type, the input is not matched. In the case of the second type, the obtained matching is the best one. The input matching of the third type depends on the size capacities of $C_{1}$ and $C_{2}$. From Figure 8a, it can be seen that the differential matching to 100Ω is almost identical in all three cases. The small differences are due to different input transistors. In the case of Type II, four times smaller transistors were used compared to Type I and Type III, which have smaller parasitic capacitances and result in better matching at higher frequencies. From the input impedance calculation and the compared characteristics, it can be seen that the emitter follower shows capacitive behavior. When using larger input transistors, the mismatch at high frequencies increases. Therefore, a choice must be made between a trade-off of transistor size, input matching, and the output current of the resulting circuit, i.e., the ability to excite the next stage.

The ability of the emitter follower to excite other circuits is predetermined by its low output resistance. Therefore, emitter followers are also used as output stages. The emitter follower is used as an “impedance transformer” with high input and low output resistance. Figure 9 shows the use of emitter followers as the output stage in differential amplifier designs.

The circuit can be powered by negative voltage determined by *VEE* and *GND*, positive voltage determined by *VCC* and *GND*, or a symmetrical voltage determined by *VCC* and *VEE*. The emitter follower is controlled by a current source consisting of NMOS transistors. The output resistance of the emitter follower on the schematic for small signals in Figure 5b if $ro\gg 1/g_{m}$ can be determined as follows [29]: (15)$$R_{o}=\frac{r_{\pi }}{1+\beta }\|R_{L},$$
where $R_{L}$ represents the resistance of the current source. Equation (Equation 15) shows that increasing the current of the current source increases the gain of the bipolar transistor $Q_{1}$ and decreases the output resistance of the transistor, $Q_{1}$ and $R_{L}$ of the current source, thus decreasing the total output resistance of the emitter follower. In our case, for 0.35 µm BiCMOS technology, using the standard NPN transistors with a current gain factor of approximately β≈200 and a current source $I_{Q}\approx 18$ mA, the emitter follower achieves an output resistance of about $R_{o}\approx 2\;\Omega$. The output stage should also provide a large output voltage swing without distortion. This means that for proper excitation of other circuits, the operating point of the emitter follower has to be set so that the output signal is not clipped by saturation of the transistors $Q_{1}$ and $M_{2}$, Figure 9.

The transmission characteristics of the emitter follower as an output stage based on the assumption $R_{s}=0\;\Omega$ are shown in Figure 10a. The emitter follower is driven by the voltage $V_{i}$, which is offset by the DC offset value *V_BE_*, so that the DC component of the output voltage $V_{o}$ is approximately at the center of the DC characteristic. From the positive part of the conversion characteristic, the output voltage is limited by the saturation of the transistor $Q_{1}$. In the negative part of the transfer characteristic, the output voltage is limited by the magnitude of the load *R* and the current $I_{Q}$ of the current source. Increasing the current $I_{Q}$ will increase the voltage swing across the load *R*, which is limited by the voltage $V_{EE}+V_{M2sat}$. On the other hand, decreasing the resistance *R* will increase the load and, thus, limit the output voltage swing, which can be seen at the bottom of the characteristic in Figure 10a.

From Equation (Equation 15) and simulations created in 0.35 µm BiCMOS technology, the output impedance of the emitter follower was determined to be of the order of Ω units. In a 50Ω environment, this output impedance is not suitable, therefore, a resistor $R_{s}$ was added to the series. For correct matching, the total output resistance must be equal:(16)$$R_{o}=\left(\frac{r_{\pi }}{1+\beta }\|R_{L}\right)+R_{s}\approx 50\;\Omega$$
The resulting conversion characteristic of the output voltage $V_{R}$ to load *R* is shown in Figure 10b. In the case, R↦∞, the output voltage $V_{o}=V_{R}$. In other cases, the output voltage $V_{R}$ is given by:(17)$$V_{R}=V_{0}\left(1-\frac{R_{s}}{R+R_{s}}\right)$$

If the total output impedance of the output is equal to the load, the resulting voltage at the load is a half. Figure 11 shows the output sinusoidal waveform with a total output impedance of 50Ω of the emitter follower with different load impedances, at a signal frequency of 1 GHz. It is also worth mentioning the output impedance of the emitter follower and the circuit behavior under capacitive loading, including the NMOS transistor in the current source. The presented output resistance of the emitter follower (Equation 16) is valid if the signal frequency ω↦0. The model for calculating the output impedance of the emitter follower for high frequencies is shown in Figure 12. The resistor $R_{S}$ represents the output impedance of the signal source.

At high frequencies, $C_{\pi }$ can short the resistor $R_{\pi }$. Neglecting the effect of the capacitance of $C_{\mu }$, we can approximate the output impedance as [29,33]:(18)$$Z_{o}=\left(\frac{R_{S}+z_{\pi }}{1+\beta }\right)\|Z_{L},\;\mathrm{where}:\;z_{\pi }=\frac{r_{\pi }C_{\pi }}{1+j\omega r_{\pi }C_{\pi }}$$

Based on the schematic in Figure 12 and the characteristic in Figure 13b, it is possible to say that the output impedance $Z_{o}$ of the emitter follower at high frequencies shows inductive behavior. Where the frequencies of $\omega_{1}$ and $\omega_{2}$ are [33]:(19)$$\omega_{1}\approx \frac{1}{r_{\pi }C_{\pi }}\;\omega_{2}\approx \beta_{0}\omega_{1}\approx \frac{\beta_{0}}{r_{\pi }C_{\pi }}\approx \frac{g_{m}}{C_{\pi }}$$An alternate schematic of the output impedance is shown in Figure 13a. If we assume that the load impedance of the emitter follower $Z_{\pi }$ is also capacitive, the following results in a parallel resonant circuit, where [33]:(20)$$R_{1}=R_{S}\;R_{2}=\frac{r_{\pi }}{1+\beta_{0}}\;L\approx \frac{g_{m}}{C_{\pi }}\left(R_{1}+R_{2}\right)\;Z_{L}=R_{L}\|\frac{1}{j\omega C_{x}},$$

where $R_{L}$ represents the DC output resistance of the current source (NMOS transistor $M_{2}$
Figure 9) and $C_{x}$ represent the parasitic capacitances of the NMOS transistor $M_{2}$ and possibly other parasitic capacitances of the circuit that are excited by the emitter follower. Figure 14a shows the simulation results in 0.35 µm BiCMOS technology according to the scenarios in Figure 14b. Three scenarios were created, where the emitter follower was loaded with an NMOS and a bipolar current source with an attached differential amplifier.

The third scenario represents an emitter follower with only an ideal current source, loaded with artificial capacitance. It can be observed that the bandwidth of the emitter follower with the NMOS current source is larger, along with the cascading of two emitter followers with the NMOS current source. With the emitter follower loaded with an ideal current source and a capacitance of 300 fF, it is possible to observe a gentle resonance and gain increase. By using emitter followers with NMOS current sources or cascading them, it is possible to increase the bandwidth of differential amplifiers. The input and output matching were simulated with SP analysis in the CADENCE Virtuoso design environment with ideal 50 Ω connected to the amplifier. This simulation was performed using RC-extract and approximate wire-bond induction. The input and output matching resistors were tuned using a Smith diagram and an S11 linear graph. The simulations were conducted at a set temperature of 27 °C. For more information about the transmission characteristics of the emitter follower, see [29,30,33].

## 5. Implemented Wideband Differential Amplifiers in 0.35 µm SiGe BiCMOS Technology

In the process of designing circuits, it is necessary to choose the input and output circuits and the amplifier concept. It is impossible to create a perfect wideband amplifier that achieves perfect bandwidth, gain, noise, power consumption, and CMRR at the same time. Therefore, several types of differential amplifiers were created, which have their own specific dominant characteristics. Differential amplifiers were primarily designed for use in a wideband directional coupler, but they have also been designed as stand-alone on-chip structures for testing purposes, as well as for other potential uses. The design requirements for wideband differential amplifiers were largely based on the requirements of the wideband directional coupler [18].


**Differential Amplifier Design Requirements:**
Gain from 10 dB to 20 dB single-ended.Amplifier frequency bandwidth from “DC” to the maximum possible, at least 10 GHz.High CMRR, of the order of tens of dB.Input differential impedance 50Ω.Output single-ended impedance 50Ω (differential 100Ω).Highest compression point P1dB (min. −15 dBm).Supply voltage −3.3 V (compatible with sensor system [20]).Low power consumption.


A total of six types of amplifiers were manufactured separately, namely, DIFF15-01, DIFF15-03, DIFF15-04, DIFF15-05, DIFF15-LN, and DIFF15-06. All of the amplifiers presented have an amplifier core consisting of a modified structure Cherry–Hooper with active feedback, which has been additionally modified.

The basic block diagram for all the designed differential amplifiers is shown in Figure 15. The DIFF15-01, DIFF15-03, DIFF15-04, DIFF15-05, and DIFF15-06 amplifiers consist of three main blocks as follows: a core consisting of a Cherry–Hooper structure and emitter followers that represent the input and output circuits. The DIFF15-LN amplifier has no input emitter followers and the input signal is fed directly to the input of the differential stage.

### 5.1. Implementing Emitter Degeneration for Bandwidth Enhancement, DIFF15-01, DIFF15-03

Although the Cherry–Hooper structure has better frequency characteristics than the classic differential amplifier, this structure was modified using emitter feedback, called emitter degeneration. There are two types of emitter degeneration wiring, namely, π-type and T-type. The π structure was used, which uses two current sources per transistor differential pair. Connecting type π has an advantage in that it does not shift the DC operating point by the voltage drop across the degenerating resistors, as is the case for the T circuit, which uses one current source and two degenerating resistors [29]. When a feedback emitter resistor is added, the linear input range of the differential well is extended by a value, which is approximately equal to $I_{T}R_{E}$, Figure 16b, where $R_{E}$ is a degradation resistor [29]. In our specific case, according to the schemes in Figure 16a and Figure 17a, $R_{E}=R_{2}$.

Based on the schematic in Figure 17 and the Equations (Equation 2)–(Equation 4), the total gain of the Cherry–Hooper differential amplifier can be determined by:(21)$$A_{d}=\frac{v_{od}}{v_{id}}=\frac{g_{m1}\left(R_{7}+R_{4}\right)\left(1/g_{m5}+R_{3}\right)}{\left(1/g_{m3}+R_{7}\right)\left(1+g_{m1}Z_{E}\right)}\;\mathrm{where}:\;Z_{E}=\frac{R_{2}}{2}\|\frac{2}{j\omega C}$$
Emitter degradation will reduce the differential gain of the amplifier, but increase the frequency bandwidth of the amplifier. By adding the capacitance *C*, which is in parallel to the resistor $R_{2}$, the impedance $Z_{E}$ starts to decrease at higher frequencies, which will again increase the gain of the amplifier. This will create what is known as “capacitive peaking”. The disadvantage of this modification is a slight degradation of the separation of the common-mode gain from the difference gain. A comparison of the measured voltage gains and the bandwidth of the amplifiers with emitter degeneration and capacitive peaking is shown in Figure 17b.

The DIFF15-01 achieves a larger frequency bandwidth relative to the DIFF15-03 amplifier. For DIFF15-01, higher values of the capacitance *C* and the resistor $R_{2}$ were used compared to the amplifier DIFF15-03. This produces larger capacitive peaking compared to the DC gain value of the amplifier. Both amplifiers achieve the largest bandwidth over 12 GHz (−3 dB), across all amplifiers. The lower DC gain in DIFF-01 is due to poor layout design. The cascading of two emitter followers at its input also contributed to the increased bandwidth for the DIFF15-01. Emitter degeneration has one more positive effect, namely, increasing the input compression point of P1dB amplifiers. In the case of DIFF15-01, this has P1dB = −8.5 dBm, and in the case of DIFF15-03, this has P1dB = −12 dBm. Figure 18 shows the two amplifiers implemented on the chip during measurements on a probe-station.

### 5.2. Using Parasitic Capacities for Bandwidth Enhancement, DIFF15-04, DIFF15-05

The Cherry–Hooper structure is a very rewarding structure for experimentation. As part of the research, it became apparent that it is possible to use the parasitic capacities of some elements to achieve a higher bandwidth. In this case, two structures, DIFF15-04 and DIFF15-05, were designed. The designs use the ratio of resistors $R_{6}\left(R_{7}\right)$ and $R_{3}\left(R_{4}\right)$, shown in Figure 19a, respectively, and the ratio of resistors $R_{1}$ and $R_{2}$, shown in Figure 2, which are part of the active feedback of the amplifier.

Applying Equation (Equation 4) to the scheme in Figure 19a, we obtain the following relation for the gain calculation:(22)$$A_{d}=\frac{v_{od}}{v_{id}}=\frac{g_{m1}\left(Z_{x}+R_{3}\right)\left(1/g_{m5}+R_{2}\right)}{\left(1/g_{m3}+Z_{x}\right)},\;\mathrm{where}:\;Z_{x}=R_{6}\|\frac{1}{j\omega C_{x}}\;$$

From the Equation (Equation 22), it can be assumed that by decreasing the impedance $Z_{x}$, the gain of the differential amplifier increases. If we assume that the impedance $Z_{x}$ consists of the resistor $R_{6}$ and the parasitic capacitances $C_{x}$, it is possible, by increasing these capacitances, to create a gain increase at higher frequencies, thus applying capacitive peaking to the gain of the amplifier. By increasing or decreasing the total area of the resistors $R_{6}\left(R_{7}\right)$ on the chip, it is possible to control this peaking.

Figure 20a shows a comparison of the amplifier measurement at the probe-station with the simulation result. The high frequency gain of the amplifier can be increased by adding an “artificial capacitance” $C_{x}$ to the ideal resistor. In the case of the DIFF15-05 amplifier, the capacitive peaking control was improved when the $R_{6}$ resistor was replaced by PMOS transistors.

The transistor operates in active mode with a control voltage $V_{GS}$ from −3.3 V to −1.8 V; higher voltages close the transistor and detune the operating set point of the amplifier. If the voltage $V_{GS}=-1.8$ V, the transistor has a low gain $g_{m}$ and, thus, higher output resistance, and the parasitic capacitance $C_{x}$ of the PMOS transistor dominates. This reduces the DC gain and creates capacitive peaking at higher frequencies. If $V_{GS}=-3.3$ V, the transistor is fully open, the output resistance of the transistor is reduced, and the resistive character dominates over the capacitive one [29]. The voltage gain of the amplifier as a function of the voltage $V_{GS}$ of the PMOS transistors is shown in Figure 20b. The default DC gain of the amplifiers was set to approximately 16 dB. Figure 21 shows the amplifiers implemented on the chip during measurements and wire-bonded to package.

### 5.3. Implementation of Low-Noise Differential Amplifier DIFF15-LN

The noise or noise figure is another important parameter, especially for amplifiers in receivers. So far, the previous designs focused on bandwidth and amplifier matching and it was found that both input matching and emitter followers were contributing to the increasing noise. Each element in the circuit is a particular source of noise. The simplified models of resistor and bipolar transistor with noise sources are shown in Figure 22a and the typical noise current spectral density waveform in bipolar circuits is shown in Figure 22b [29].

The flicker noise 1/f is also known as pink noise, which is found in all active devices, as well as some discrete passive elements, such as carbon resistors. Thermal noise is generated by a completely different mechanism. For example, in conventional resistors, it is generated due to random movement of electrons, due to the thermal conditions. The shot noise is always associated with DC current and is present in diodes, MOSFETs, and bipolar transistors. It is formed by the transition of charge carriers through a semiconductor junction. High-frequency noise $f^{2}$ is caused by decreasing gain of the transistor, resulting in a gradual degradation of the transistor’s noise performance observed at high frequencies. Based on the complex noise analysis presented in [29], to achieve lower noise in the amplifier, it is necessary to reduce the number of components, reduce the internal resistances of transistors, increase the current gain as much as possible, and reduce the current of the bipolar transistors. It is necessary, therefore, to increase as much as possible the current gain $\beta_{0}$ and to reduce the current $I_{c}$ of the bipolar transistors. A schematic diagram of the DIFF15-LN wideband differential amplifier with input circuits is shown in Figure 23.

The largest noise contribution is produced by the input matching resistor $R_{1}$, which is connected directly to the differential input, but without this resistor, the input would not be matched. According to the principle of reducing the number of components, the input emitter followers were removed. The emitter followers have no voltage gain at the input, and they contribute to the overall noise floor of the amplifier. In the case of low-noise amplifiers, the idea is to amplify the input signal directly in the first stage. For the design of the amplifiers presented in this paper, NPN232 transistors are used, with two emitters and collectors and three bases. These transistors have the highest $f_{t}$ within the 0.35 µm SiGE BiCMOS technology used. For the DIFF15-LN amplifier, larger NPN243 transistors with an emitter length of 48 µm with two collectors, four bases, and three emitters were used. Larger transistors provided smaller internal resistances, higher $\beta_{0}$, and current density at the same current as in the case of DIFF15-04. The disadvantage of using larger transistors is that the bandwidth of the amplifier is reduced.

It can be seen that DIFF15-01 has the highest noise figure, due to the use of cascaded emitter followers at the input. The second highest noise figure, due to the use of the smaller transistor NPN232 with an emitter length of 24 µm, is the DIFF15-06, which reduced $\beta_{0}$ and degraded the transistor size ratio to the collector current $I_{c}$. DIFF15-04 uses NPN232 transistors with emitter length 48 µm with the same collector current $I_{c}$ as the DIFF15-LN, which also has the lowest noise figure. Figure 24b shows the input matching of the DIFF15-LN amplifier measured by the probe-station and compared with the simulated RC extract of the DIFF15-LN amplifier and the measured input matching of the DIFF15-04 amplifier. It can be observed that the DIFF15-04, which has emitter followers at the input, shows better input matching.

### 5.4. CMRR Optimization, DIFF15-06

The DIFF15-06 was the last of the proposed amplifiers that was designed. Based on a backward analysis of the previously designed amplifiers, it was found that the CMRR parameter was insufficient. Therefore, the goal of this design was to improve the CMRR and reduce the overall circuit power consumption. The core of the DIFF15-06 amplifier is shown in Figure 25a. As can be seen, the reference resistor was replaced by an NMOS transistor according to the concept presented in Figure 3b. The reference resistor had a high resistance and used a lot of the area on the chip; by replacing it with an NMOS transistor space was saved. Figure 25b shows an alternative schematic to explain the CMRR improvement of the differential amplifier. As is known, the higher the internal resistance of the current source $R_{T}$ of the differential amplifier that is achieved, the higher the common-gain reduction and increase in the CMRR.

A current source consisting of transistor $M_{1}$ has higher values of internal resistance if its voltage $V_{DS}>V_{sat}$, with $V_{sat}$ the saturation voltage of the NMOS transistor. In the case of the previous amplifiers, it was found that the output voltage was set to a high output voltage swing, which was limited by the supply voltage. The voltage drops in the circuit caused the $V_{DS}$ voltage to attach the limit $V_{sat}$ in some cases. Another problem was the high saturation voltage of the NMOS transistors of the current source. Based on Equation (Equation 8), the saturation voltage $V_{sat}$ can be reduced by increasing the width of the NMOS transistor while keeping the same output current $I_{T}$. In the case of the DIFF15-06, the NMOS transistors are almost double the width of the previous designs. In order to keep the same current $I_{T}$, the voltage $V_{GS}$ was reduced. This improved the suppression of the common-gain at lower frequencies, as shown in Figure 26b.

Figure 25b shows the influence of the voltage $V_{GS}$ on the voltage $v_{gs}$. When the common-mode signal is connected, the voltage $v_{ds}$ almost follows this voltage $v_{1}$. Part of this voltage is applied through the capacitance $C_{gd}$ to the gate, which causes instability of the current source at high frequencies. In the case of the amplifier common-mode excitation by the normalized amplitude of 1 V, the crosstalk of the $v_{ds}$ voltage to the $v_{gs}$ voltage was up to 82 mV. Adding the capacitance *C* creates a capacitive divider and the voltage $v_{gs}$ is almost eliminated. Figure 26a shows a simulation of the suppression of the voltage $v_{gs}$ at the gate of the transistor. Within the chip, a capacitance of C=8.5 fF was implemented. For possible connection of external capacitances outside of the chip, the bonding *BIAS* contact was added.

Figure 26b shows the resulting suppression of the common-mode gain depending on the capacity *C*. Worse common-mode gain suppression at frequencies above 10 GHz is caused by parasitic capacitances $C_{s}$ and $C_{db}$, where $C_{s}$ represents the direct capacity to the ground and $C_{db}$ is one of the parasitic capacitances of the NMOS transistor. The capacitances reduce the resulting impedance of the current source at high frequencies, which leads to worse CMRR. The final CMRR in decibels is the difference between the differential gain and the common gain. The resulting comparison of the CMRR, for schematic and RC extract simulation of the DIFF15-06 and DIFF15-04 amplifiers, is shown in Figure 27a. It can be seen that the CMRR reaches up to 70 dB. Figure 27b shows the DIFF15-06 differential amplifier wire-bonded in a QFN32 package with additional circuits on a single chip.

## 6. Evolution of Layout and Comparison

With the development of the schematic circuits of amplifiers, the layout of the elements on the chip, i.e., the layout of the individual parts of the amplifiers, also evolved. Figure 28 shows detailed layouts of the presented differential amplifiers, designed in the CADENCE Virtuoso design environment. The view is to a uniform scale for better interpretation of the amplifier size. The amplifiers were designed and fabricated in 0.35 µm SiGe BiCMOS technology from the Austrian manufacturer AMS.

In the layout (Figure 28), a reduction in the area occupied by individual cells is visible. The amplifiers were designed with structures as symmetrical as possible to keep the best differential parameters. The DIFF15-01 amplifier was the largest, with more than twice the surface area of the DIFF15-06 amplifier. The biggest problems with the DIFF15-01 amplifier were high power consumption and miscalculation of the current density of metallic interconnects and the vias between the layers, and longer leads due to the cell size. This caused a reduction in the low-frequency gain. The amplifier was designed for a gain of 15 dB, but only achieved 12 dB.

The overall power consumption and cell size were reduced by removing the cascade of the two emitter followers. For the other designs, wider metallic interconnections were used, and they were drawn parallel to each other on two layers. The vias between the layers were increased. Additionally, reducing the size of the resistors to the edge of the current density also contributed to a reduction in the amplifier cell size. In 0.35 µm BiCMOS S35D4M5 technology, the minimum length/width ratio of resistors is 5:1.

The smaller resistor area also achieved a reduction in the parasitic capacitance, which contributed to increased bandwidth and better input-output matching at higher frequencies. Some types of resistors offer a double row of connection via, to reduce the transient resistance and more precise adjustment. The wider and shorter signal junctions between transistors were used which, in the context, represents a lower transient resistance of the individual interconnections. This resistance connection with the parasitic capacitance represents a low-pass filter. The latest design of the DIFF15-06 amplifier surrounds power capacitances that help bypass current spikes, and has capacitances for the current source that improve CMRR.

The presented characteristics are the result of measuring the amplifiers using a probe-station. A block diagram and photo of the probe-station measurement setup are shown in Figure 29. For S-parameter probe-station measurement, an Agilent PNA-X N5241A vector analyzer with maximum frequency 13.5 GHz was used. Two types of micro-probes, namely, a CASCADE MICTROTECH ACP40-AW-SG-100 with 40 GHz bandwidth and an MPI TITAN T26V-SG0100 probe with 26 GHz bandwidth, were used. Because the probes are only in an SG (signal–ground) configuration, a quasi-differential connection was applied with a PE8212 inner outer DC-block capacitor so that the probe could be connected to the differential input with 50 Ω. At the output, only a standard DC-block capacitor was used. The second output of the amplifier was unconnected and unmatched due to space limitations and no possibility to connect another probe. Such a probe-station best describes the actual parameters of the amplifiers as they are not affected by wire-bonding, packages, and PCBs. The compression point and output voltage swing was measured on a wire-bonded and assembled amplifier on PCB. A block diagram and photo of the PCB measurement setup are shown in Figure 30. For compression measurement, a signal generator Keysight N5183B and an Agilent N9020A spectrum analyzer up to 26.5 GHz were used. In the case of voltage swing measurement, the spectrum analyzer was replaced by a wideband oscilloscope Agilent DSO9404A with 20 GHz sampling. For the DC-block, on board wideband 520L103KT16T 100 nF ceramic capacitors were used. In this case, the second output was terminated with a 50 Ω load; the second input was terminated directly to the ground via a wideband ceramic capacitor. The resulting comparison of all the parameters of the proposed amplifiers is presented in Table 1.

In Figure 31a,b and Figure 32a, the measured characteristics of gain, input matching, and output matching are presented to determine the frequency bandwidth. As part of the design of amplifiers, it is important to realize that it is necessary to adjust the resulting voltage gain of all amplifiers to 6 dB higher because on a matched load, the output amplitude of the signal will be a half. This means that for a 15 dB voltage gain, it was necessary to adjust the amplifier core to be 6 dB higher, up to 21 dB. The DIFF15-01, DIFF15-03, and DIFF15-05 amplifiers have the highest bandwidth and show the most significant capacitive peaking. The lowest power consumption comes with DIFF15-06, which has smaller 24 µm NPN232 transistors and a smaller source current; on the other hand, smaller transistors also result in higher noise. If amplifiers are part of other circuits and the output would not come out of the chip, the amplifiers would not need such a large output stage, which, on average, accounts for up to 40% of an amplifier’s total power consumption. The designed amplifier assembled on the evaluation board is shown in Figure 32a. The dimensions of the fabricated chips are 2 × 2 mm, including other circuit structures which were implemented on the chip dies. A view of the fabricated chips with the amplifiers marked is shown in Figure 32b.

## 7. Conclusions

Several wideband amplifiers were designed, manufactured, and tested, which can be used in various applications with an M-sequence based radar system. The presented amplifiers were improved with each new design. With these designs, low-cost 0.35 µm BiCMOS technology was used at the limit of its capabilities. Overall, the input and output matching achieved good results, in most of the frequency range staying below 10 dB. Worse output matching observed for DIFF15-03 and DIFF15-LN was due to the way the output signal was laid out across the entire length of the chip (Figure 32b). Other amplifiers were implemented in the corners of the chip, which reduced the length of the bond wires. In terms of compression points, the highest input compression points were achieved by amplifiers DIFF15-01 and DIFF15-03 with emitter degeneration. The DIFF15-06 amplifier had the best overall performance; it achieved a large frequency bandwidth, with low power consumption and a high CMRR. The only disadvantage was the noise figure for this amplifier. The amplifiers were wire-bonded in QFN24 and QFN32 packages and assembled on development boards, as shown in Figure 33.

## Figures and Tables

**Figure 1 sensors-24-02143-f001:**
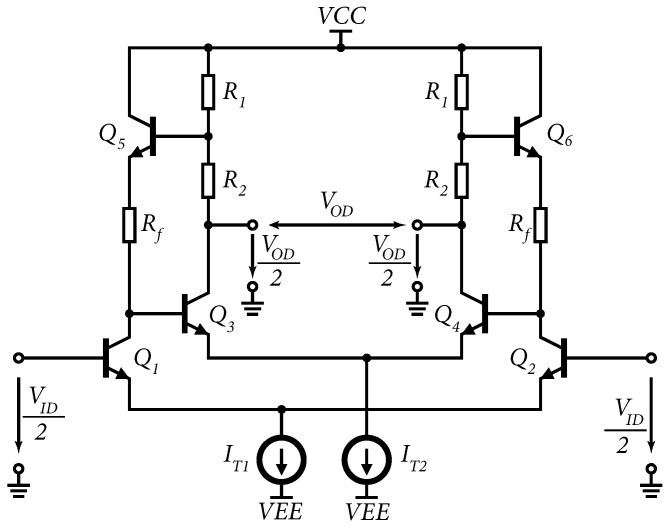
Modified structure of Cherry–Hooper amplifier with feedback created using emitter followers formed by transistors $Q_{5}$ and $Q_{6}$.

**Figure 2 sensors-24-02143-f002:**
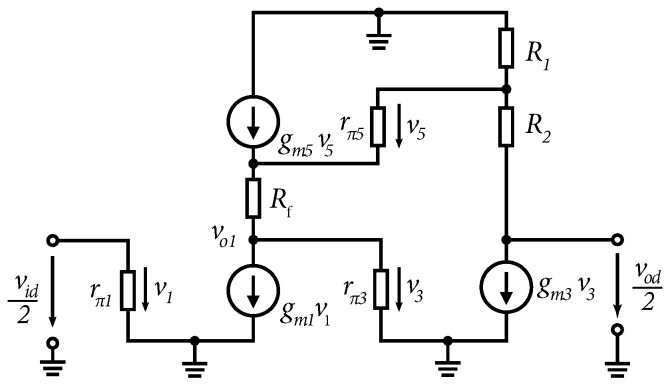
Equivalent half-circuit diagram for signals of modified Cherry–Hooper structure with transistor model substituted.

**Figure 3 sensors-24-02143-f003:**
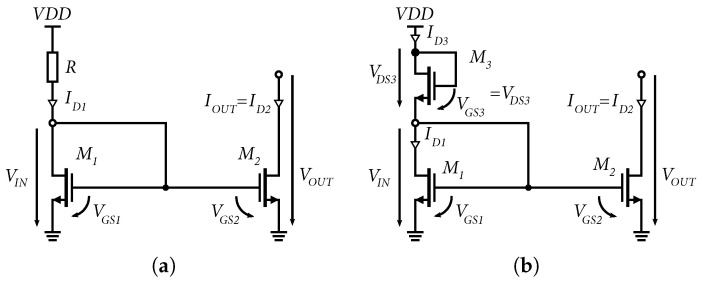
Current mirror schematics with NMOS transistors. (**a**) Standard NMOS current mirror schematic. (**b**) Full NMOS current mirrors chematic.

**Figure 4 sensors-24-02143-f004:**
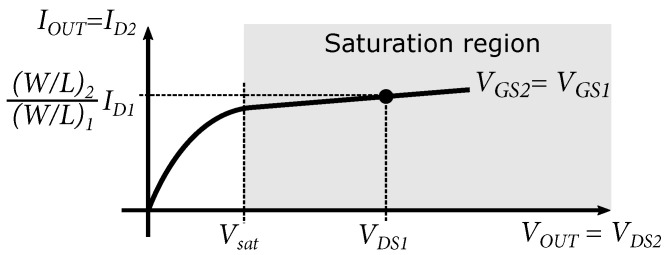
Output characteristic of current mirror with NMOS transistors.

**Figure 5 sensors-24-02143-f005:**
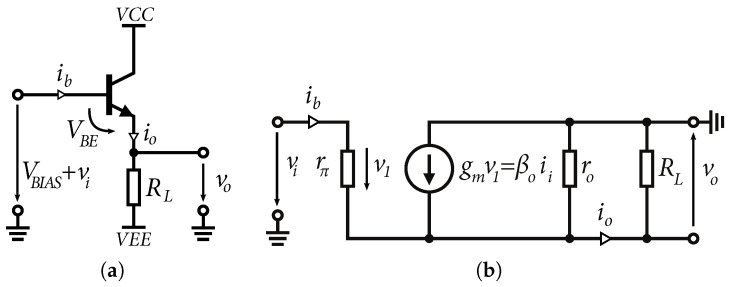
Basic and small-signal model of emitter follower. (**a**) Basic schematic of the emitter follower. (**b**) Equivalent small-signal model of emitter follower.

**Figure 6 sensors-24-02143-f006:**
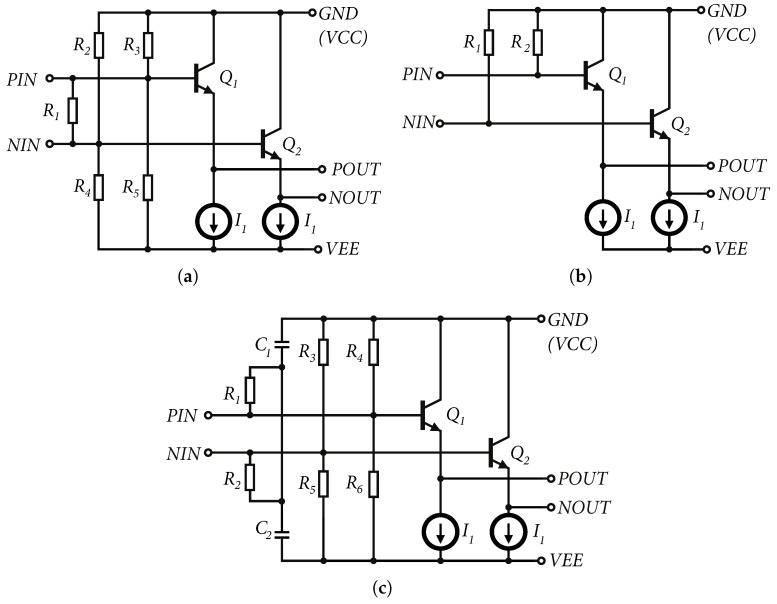
Wideband amplifiers described in this article (**a**) Input circuits Type I. (**b**) Input circuits Type II. (**c**) Input circuits Type III.

**Figure 7 sensors-24-02143-f007:**
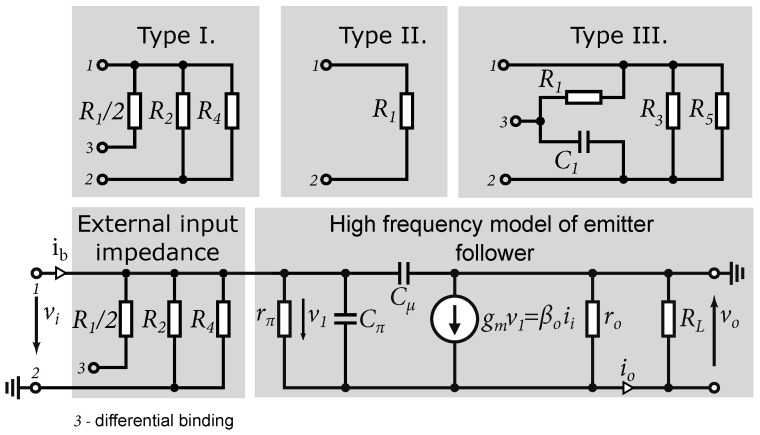
The small-signal equivalent input circuits.

**Figure 8 sensors-24-02143-f008:**
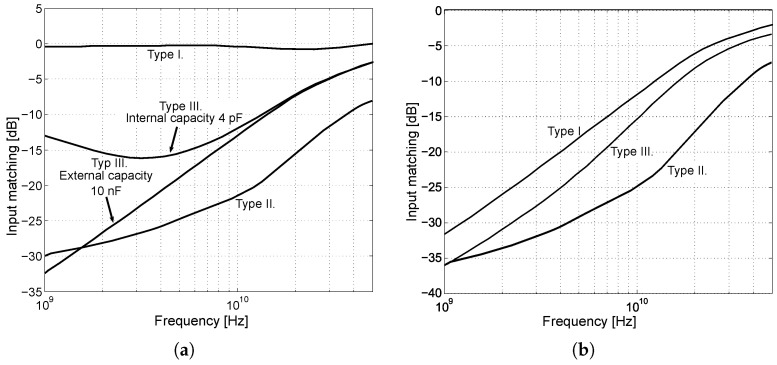
Input matching of individual types of input circuits. (**a**) Single-ended matching. (**b**) Differential matching.

**Figure 9 sensors-24-02143-f009:**
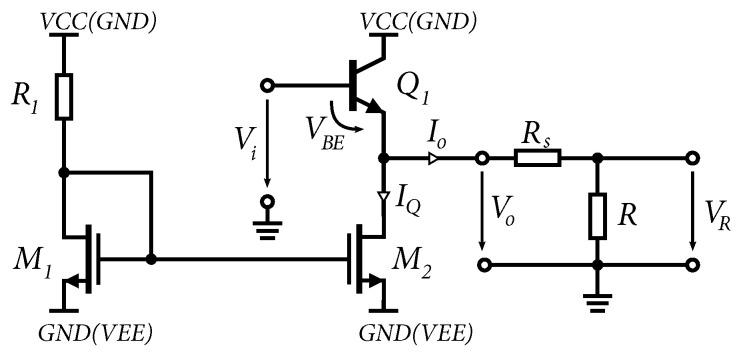
An emitter follower connected as an output stage including a load *R*.

**Figure 10 sensors-24-02143-f010:**
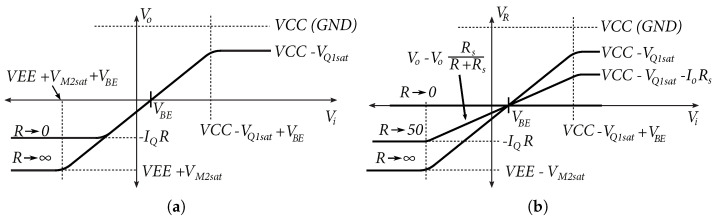
Output characteristics of the emitter follower. (**a**) Output characteristic of the emitter follower if $R_{S}=0$. (**b**) Output characteristic of the emitter follower with $R_{S}$.

**Figure 11 sensors-24-02143-f011:**
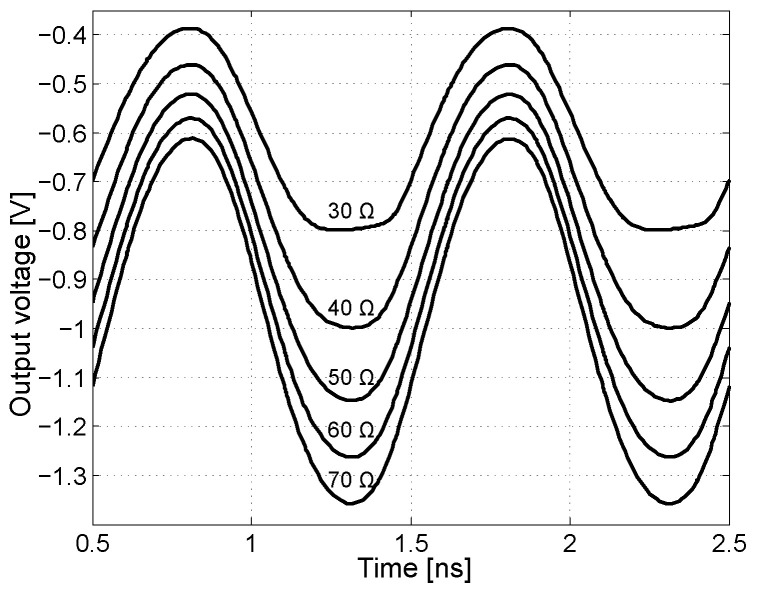
Measurement of emitter follower as output stage with 50 Ω output impedance with connection of different values of loads, at a signal frequency of 1 GHz.

**Figure 12 sensors-24-02143-f012:**
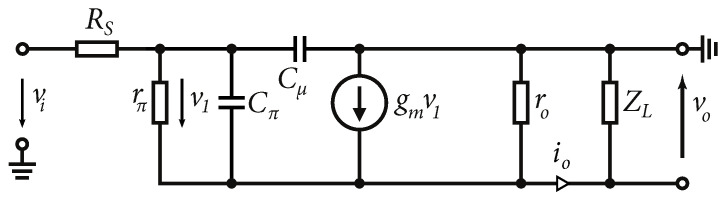
Small-signals model to determine output impedance, including high-frequency characteristics.

**Figure 13 sensors-24-02143-f013:**
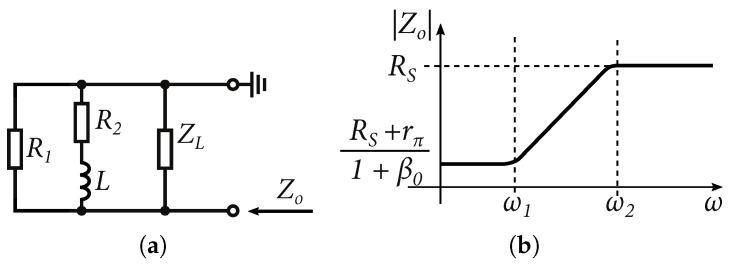
(**a**) Alternate shematic of emitter follower output impedance (**b**) Output impedance of emitter follower versus frequency [33].

**Figure 14 sensors-24-02143-f014:**
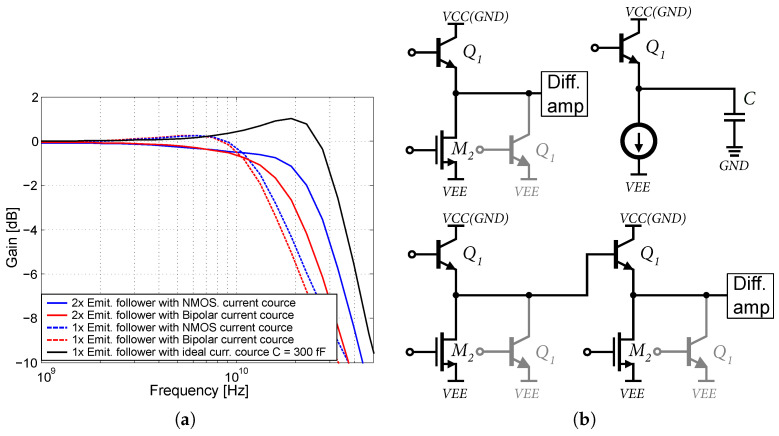
Emitter follower output bandwidth simulations and simulation scenarios. (**a**) Simulation of emitter follower output bandwidth versus load capacitance. (**b**) Emitter follower output bandwidth simulation scenarios.

**Figure 15 sensors-24-02143-f015:**
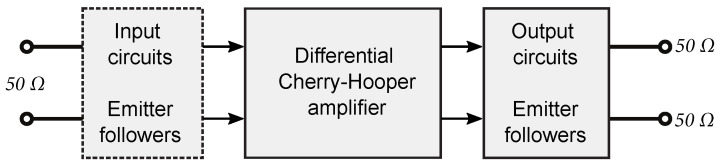
Block diagram schematic of differential amplifier structure.

**Figure 16 sensors-24-02143-f016:**
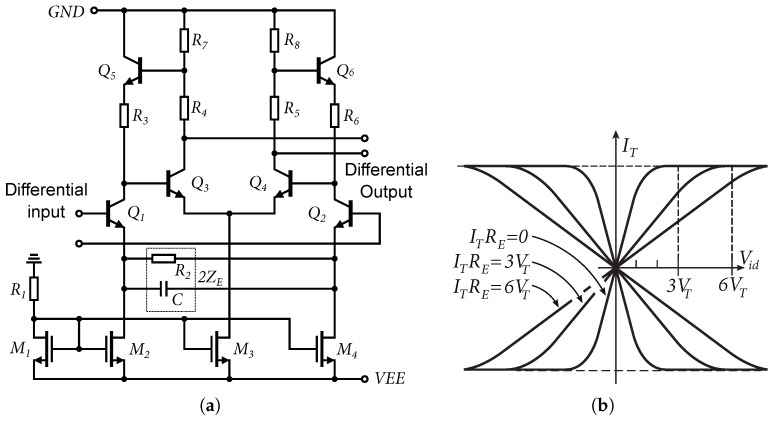
Differential core and output current with emitter degeneration. (**a**) Differential of Cherry–Hooper amplifier core used for DIFF15-01 and DIFF15-03 amplifiers. (**b**) Output current of a differential amplifier as a function of input voltage with emitter degeneration.

**Figure 17 sensors-24-02143-f017:**
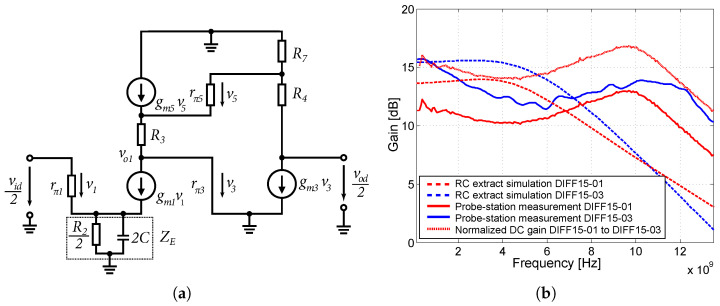
Half-circuit and comparison of the voltage gains. (**a**) Half-circuit schematic for small signals of modified Cherry–Hooper structure with emitter degeneration. (**b**) Comparison of the voltage gains of DIFF15-01 and DIFF15-03 amplifiers with emitter degeneration and capacitive peaking.

**Figure 18 sensors-24-02143-f018:**
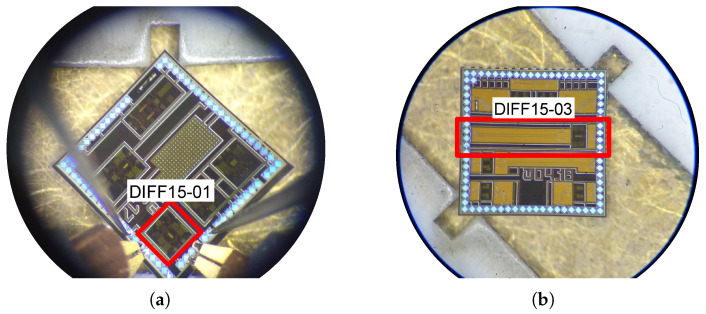
Demonstration of the fabricated chips of the designed amplifiers during the measurement at the probe-station. (**a**) Die with DIFF15-01 marked amplifier cell. (**b**) Die with DIFF15-03 marked amplifier cell.

**Figure 19 sensors-24-02143-f019:**
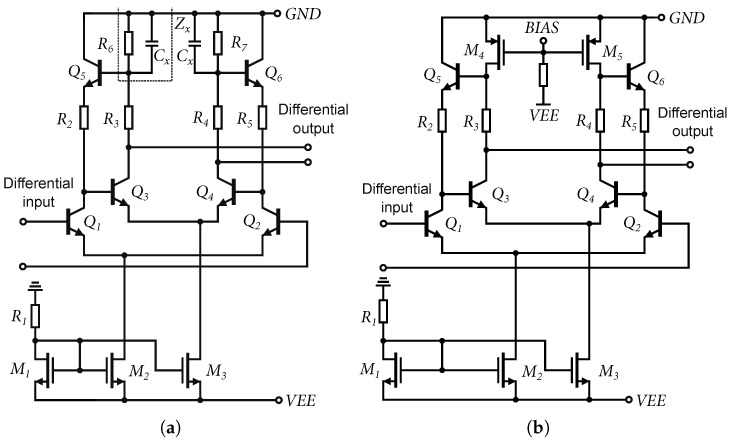
Schematics of DIFF15-04 and DIFF15-05 amplifiers core. (**a**) Schematic of DIFF15-04 amplifier core. (**b**) Schematic of DIFF15-05 amplifier core.

**Figure 20 sensors-24-02143-f020:**
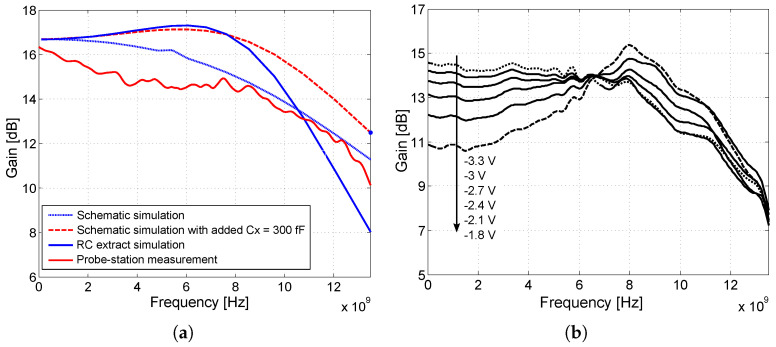
Simulations and probe-station measurements on the DIFF15-04 and DIFF15-05 amplifiers. (**a**) Comparison of simulations and resulting measurements on probe-station, amplifier DIFF15-04. (**b**) Measured dependence of gain and bandwidth at different values of voltage $V_{GS}$ of PMOS, amplifier DIFF15-05.

**Figure 21 sensors-24-02143-f021:**
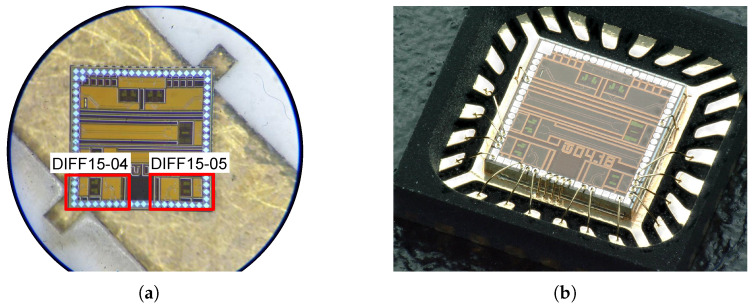
The dies of DIFF15-04 and DIFF15-05 amplifiers. (**a**) Amplifiers during probe-station measurements. (**b**) Wire-bonded amplifiers in QFN24 package.

**Figure 22 sensors-24-02143-f022:**
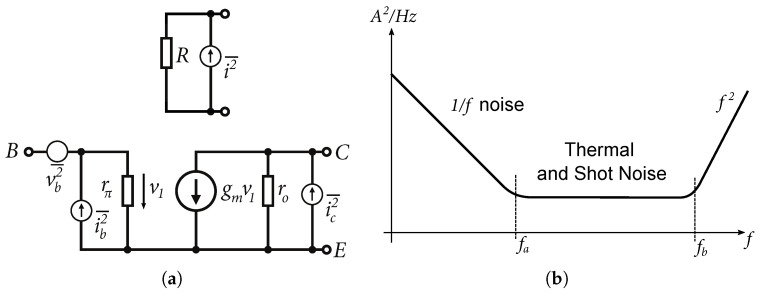
Alternative schematics with noise sources and noise current spectral density. (**a**) Alternative schematic of a bipolar transistor with internal noise sources. (**b**) Typical noise current spectral density waveform in bipolar circuits.

**Figure 23 sensors-24-02143-f023:**
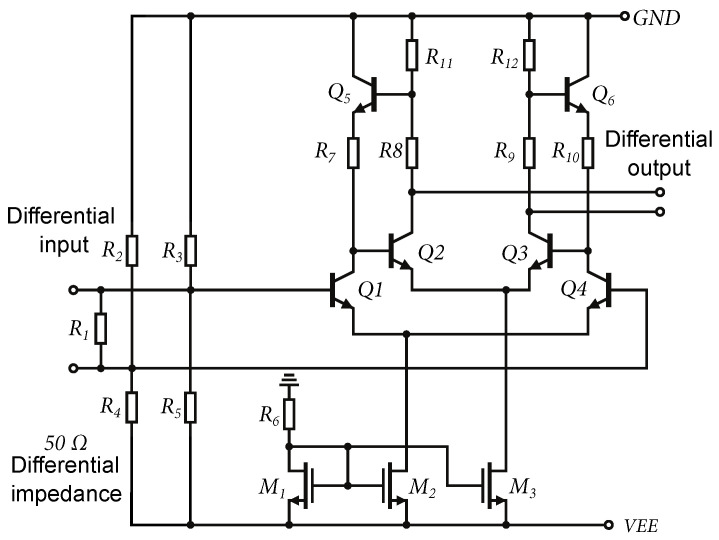
Schematic of DIFF15-LN wideband differential amplifier with input circuits.

**Figure 24 sensors-24-02143-f024:**
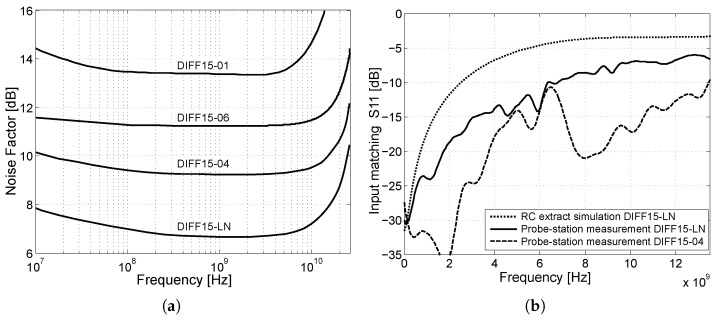
Noise figure simulation and matching measurements on the probe-station. (**a**) Noise figure comparison of the proposed amplifiers. (**b**) Measured input matching of the DIFF15-LN amplifier.

**Figure 25 sensors-24-02143-f025:**
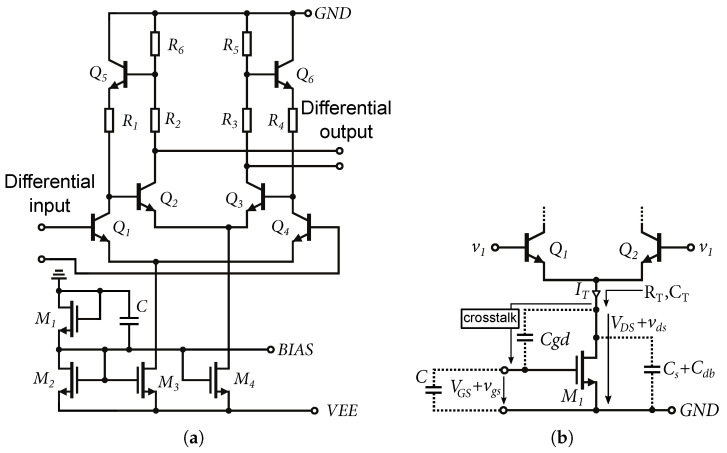
Core of DIFF15-06 and CMRR enhancement schematic. (**a**) Core of DIFF15-06 amplifier. (**b**) Alternate schematic to explain CMRR enhancement.

**Figure 26 sensors-24-02143-f026:**
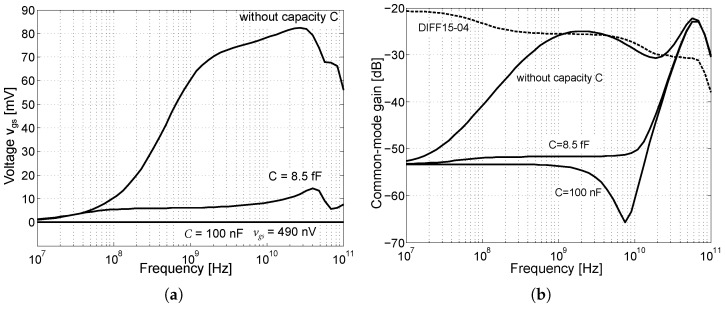
Simulation of the DIFF15-06 amplifier to determine the effect of $v_{ds}$ on the gate of an NMOS transistor. (**a**) Voltage crosstalk $v_{ds}$ at the gate of an NMOS transistor depending on capacitance C. (**b**) Suppression of common-gain depending on capacity C.

**Figure 27 sensors-24-02143-f027:**
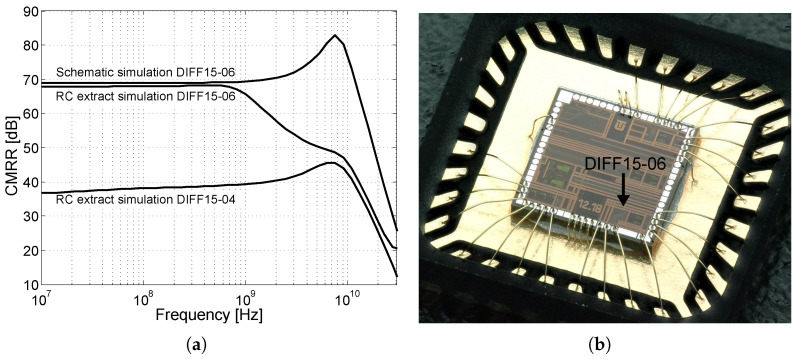
The resulting comparison of the CMRR and the encapsulated amplifier in the QFN32 package. (**a**) CMRR simulation and comparison. (**b**) Photo of wire-bonded DIFF15-06 amplifier in QFN32 package.

**Figure 28 sensors-24-02143-f028:**
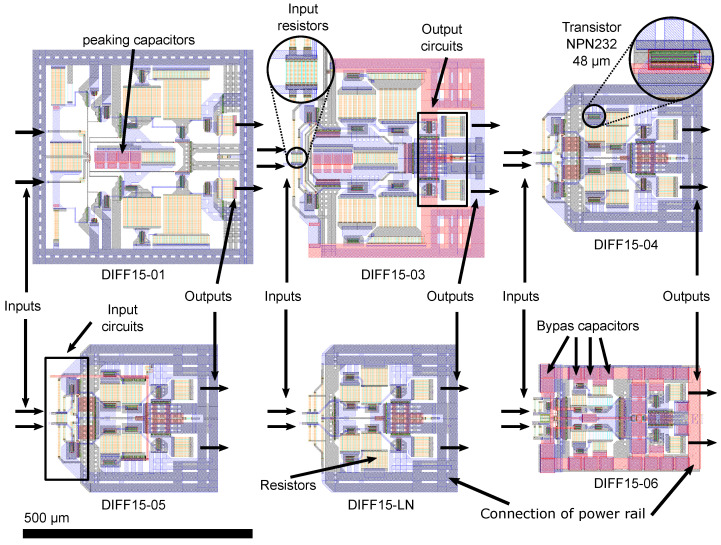
Comparison layouts of designed amplifiers.

**Figure 29 sensors-24-02143-f029:**
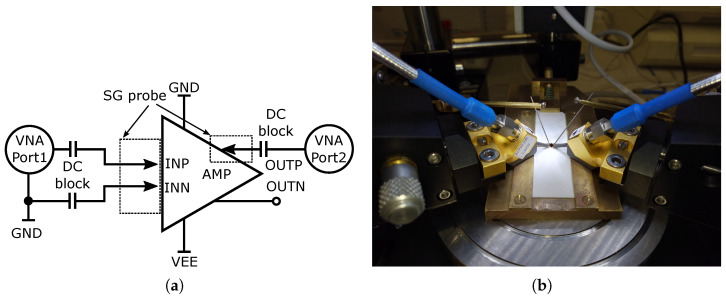
Probe-station S-parameter setup. (**a**) Block diagram of probe-station S-parameter measurement setup. (**b**) Photo of probe-station S-parameter setup.

**Figure 30 sensors-24-02143-f030:**
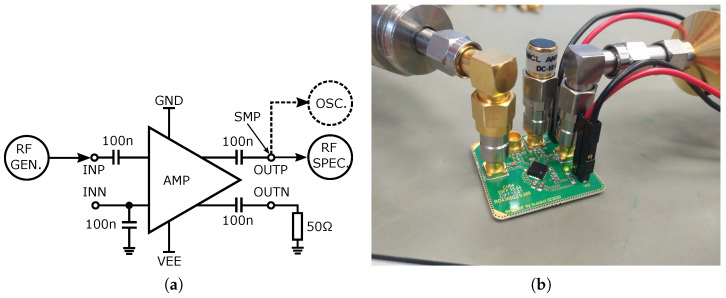
PCB measurement setup. (**a**) Block diagram of PCB measurement setup. (**b**) Photo of PCB measurement setup.

**Figure 31 sensors-24-02143-f031:**
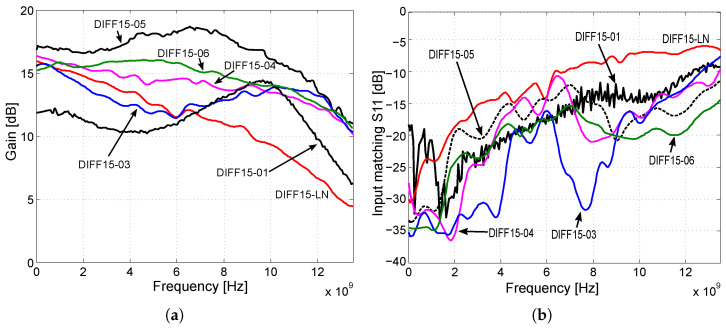
Comparison of the parameters of the designed amplifiers. (**a**) Comparison of gain as a function of frequency. (**b**) Comparison of input matching as a function of frequency.

**Figure 32 sensors-24-02143-f032:**
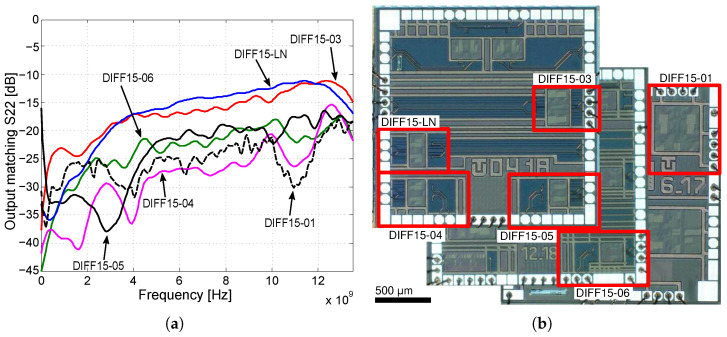
Comparison of the parameters of the designed amplifiers and display of their structure on the die. (**a**) Comparison of output matching as a function of frequency. (**b**) Designed amplifiers on the die.

**Figure 33 sensors-24-02143-f033:**
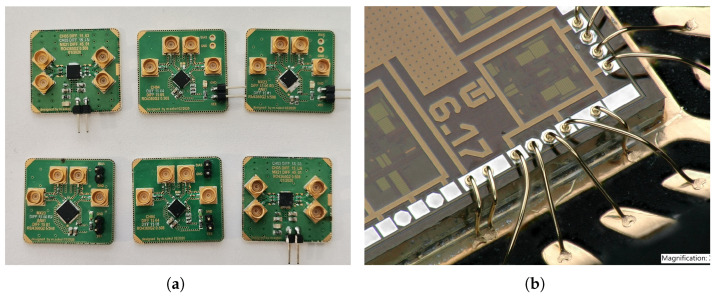
Photo of designed amplifiers assembled on PCB and wire-bond detail. (**a**) Designed amplifiers assembled on evaluation boards. (**b**) Wire bond detail of amplifier.

**Table 1 sensors-24-02143-t001:** Comparison of manufactured differentials amplifiers presented in this article.

Parameter	DIFF15-01	DIFF15-03	DIFF15-04	DIFF15-05	DIFF15-LN	DIFF15-06	[11]
Technology	SiGe 0.35 µm	SiGe 0.35 µm	SiGe 0.35 µm	SiGe 0.35 µm	SiGe 0.35 µm	SiGe 0.35 µm	SiGe 0.13 µm
Cell dimensions	490 × 480 µm	445 × 431 µm	330 × 362 µm	330 × 362 µm	330 × 337 µm	245 × 376 µm	93 × 42 µm
Supply voltage	−3.3 V	−3.3 V (3.3 V)	−3.3 V (3.3 V)	−3.3 V (3.3 V)	−3.3 V (3.3 V)	−3.3 V (3.3 V)	1.8 V
Power consumption	561 mW	480 mW	425 mW	416 mW	346 mW	343 mW	82.8 mW
Gain ($S_{21}$) 1 GHz ^3^	12 dB	15 dB	16 dB	17 dB	16 dB	16 dB	2.5–16 dB
Bandwidth (−3 dB) ^3^	12 GHz	12 GHz	11 GHz	12 GHz	6 GHz	11 GHz	4.3–7 GHz
Noise Figure [1 GHz] ^2^	13.2 dB	10 dB	9.5 dB	14.7 dB	6.9 dB	11.2 dB	NA
$S_{11}$ [1 GHz] ^3^	−27 dB	−33 dB	−32 dB	−32 dB	−24 dB	−32 dB	NA
$S_{22}$ [1 GHz] ^3^	−20 dB	−30 dB	−32 dB	−39 dB	−24 dB	−36 dB	NA
CMRR [1 GHz] ^1^	35 dB	39 dB	39 dB	38 dB	34 dB	59 dB	NA
P1dB [1 GHz] ^2^	−8.5 dBm	−12 dBm	−13.5 dBm	−12.5 dBm	−13.5 dBm	−12.4 dBm	−10 dBm
Max. out. voltage swing [1 GHz] ^2^	850 mVpp	820 mVpp	780 mVpp	800 mVpp	850 mVpp	700 mVpp	400 mVpp
Efficiency PAE ^4^	0.16%	0.19%	0.21%	0.22%	0.26%	0.26%	NA

^1^ Simulation results. ^2^ PCB measurements. ^3^ Probe-station measurements. ^4^ Power Added Efficiency (PAE) at −10 dBm input signal, the ratio of added RF power (RF output power minus RF input power) to DC power.

## Data Availability

No new data were created or analyzed in this study. Data sharing is not applicable to this article.

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
