# Peer review of "Design and Realization of Ultra-Wideband Differential Amplifiers for M-Sequence Radar Applications"

_sensors, 2024, doi:10.3390/s24072143_

Round 1
Reviewer 1 Report
Comments and Suggestions for Authors
The background theory and design choices are thoroughly explained. A large amount of test vehicles support the theoretical part. I do not have any significant criticism on the validity of the content or on the organization of the material.

There are several typos and errors, often of minor entity. However, their number is nonnegligible in my opinion, especially in the second part of the manuscript. Please see the attached PDF for details.
Author Response
RESPONSE TO REVIEWER COMMENTS
Manuscript Number : sensors- 2878182
Article Type : Full Length Article
Article Title : Design and Realization of Ultra-wideband Differentials
Amplifiers for M-Sequence Radars Applications
We sincerely thank the Editor and Reviewers for their valuable comments and suggestions to enhance the quality of our manuscript. We have addressed all the comments and suggestions given by the reviewers and incorporated the same in the revised manuscript.
REVIEWER #1
Comments on the manuscript entitled
The background theory and design choices are thoroughly explained. A large amount of test vehicles support the theoretical part. I do not have any significant criticism on the validity of the content or on the organization of the material.
Comment 1.: There are several typos and errors, often of minor entity. However, their number is nonnegligible in my opinion, especially in the second part of the manuscript. Please see the attached PDF for details.
Response 1.: Thank you very much for your suggestions to improve our article. We have included your feedback in the article.

Reviewer 2 Report
Comments and Suggestions for Authors
In this manuscript, the development of an ultra-wideband amplifier by using the the 0.35 μm BiCMOS technology. The device exhibits low power consumption and high CMMR.
The following first lines of the Abstract
“Amplifying wideband radio frequency and microwave signals is a fundamental part of almost every high-frequency circuit and device whether in commercial operation or laboratory conditions. Ultra-wideband (UWB) sensor applications require hardware designed directly for their specific application. The article presents the analysis, design and implementation of ultra- wideband differential amplifiers for M-sequence-based UWB applications. The differential amplifiers are designed based on the Cherry-Hooper structure and are implemented in low-cost 0.35 μm SiGe BiCMOS technology.”
repeat identically large portions of the Abstract of reference [30]. The present manuscript is a different work than [30] so please modify the Abstract in order to clearly avoid any suspicion of self-plagiarism
Fig. 14a shows nice plots of Gain versus frequency simulated for several types of current sources. Could the authors show also measured data on this ?
Author Response
RESPONSE TO REVIEWER COMMENTS
Manuscript Number : sensors- 2878182
Article Type : Full Length Article
Article Title : Design and Realization of Ultra-wideband Differentials
Amplifiers for M-Sequence Radars Applications
We sincerely thank the Editor and Reviewers for their valuable comments and suggestions to enhance the quality of our manuscript. We have addressed all the comments and suggestions given by the reviewers and incorporated the same in the revised manuscript.
REVIEWER #2
In this manuscript, the development of an ultra-wideband amplifier by using the the 0.35 μm BiCMOS technology. The device exhibits low power consumption and high CMMR.
Comment 1.: The following first lines of the Abstract
“Amplifying wideband radio frequency and microwave signals is a fundamental part of almost every high-frequency circuit and device whether in commercial operation or laboratory conditions. Ultra-wideband (UWB) sensor applications require hardware designed directly for their specific application. The article presents the analysis, design and implementation of ultra- wideband differential amplifiers for M-sequence-based UWB applications. The differential amplifiers are designed based on the Cherry-Hooper structure and are implemented in low-cost 0.35 μm SiGe BiCMOS technology.”
repeat identically large portions of the Abstract of reference [30]. The present manuscript is a different work than [30] so please modify the Abstract in order to clearly avoid any suspicion of self-plagiarism
Response 1.: Thank you very much for your suggestion, we appreciate your feedback and we decided to improve the abstract.
Abstract:
Amplification of wideband high-frequency and microwave signals is a fundamental element within every high-frequency circuit and device. Ultra-wideband (UWB) sensor applications are using circuits designed directly for their specific application. The article presents the analysis, design and implementation of ultra-wideband differential amplifiers for M-sequence-based UWB applications. The designed differential amplifiers are based on the Cherry-Hooper structure and they are implemented in low-cost 0.35 μm SiGe BiCMOS semiconductor process. The article presents an analysis and realization of several designs focused on different modifications of the Cherry-Hooper amplifier structure. Proposed amplifier modifications are focused on achieving the best result in one main parameter's performance. There are described amplifier designs modified by capacitive peaking to achieve the largest bandwidth, amplifiers with the lowest possible noise figure and designs focused on achieving the highest common mode rejection ratio (CMRR). The layout of all differential amplifiers was created and the chip was manufactured and wire-bonded to the QFN package, for evaluation purposes the high-frequency PCB board was designed. The schematic simulations, post-layout simulations and measurements of individual parameters of the designed amplifiers were performed, as well. The designed and fabricated ultra-wideband differential amplifiers have the following parameters, supply current of 100-160 mA at -3.3 V or 3.3 V, bandwidth from 6 to 12 GHz, Gain (at 1 GHz) from 12 to 16 dB, noise figure from 7 to 13 dB and common mode rejection ratio up to 70 dB.
Comment 2.: Fig. 14a shows nice plots of Gain versus frequency simulated for several types of current sources. Could the authors show also measured data on this ?
Response 2.: Thank you for your suggestion to improve our article. Unfortunately, we have only the simulations, standalone emitter follower was not manufactured, and that was not possible to measure it.

Reviewer 3 Report
Comments and Suggestions for Authors
This work presents 6 UWB differencial amplifiers for applications in radars.
The 6 designs were fabricated and tested.
Nonetheless, there are few flaws in the manuscript that must be fixed before recommend the work for publication on these transactions.
The core architecture is the Cherry-Hooper structure, which is resonably described.
On line 88, what is meant with "this chapter"?
The variables on figure 2 must be better associated to their equivalent counterparts in figure 1 (Vid/2 versus Vi1 and Vi2).
The kappa parameter in line 127 must be clarified because the documentation in PDK does not provide a closed nunmber. Therefore, since this value was taken from a textbook to get the first kick towards the theoretical design and trim the W/L dimensions with simulations, thus, this must be clarified/described in the main body of manuscript.
The authors done simulations for several Vdd and Vcm to access the reliability of the current mirrors and differencial amplifiers?
In line 143 it is stated the room temperature of 25 °C. Does the authors done simulations for other temperatures?
In equation (12) and line 188 it is stated a high input resistance. How high it is?
The experimental details and photos of setups that allowed to obtain the 6 sets of results in table 1 must be provided and explained.
Author Response
RESPONSE TO REVIEWER COMMENTS
Manuscript Number : sensors- 2878182
Article Type : Full Length Article
Article Title : Design and Realization of Ultra-wideband Differentials
Amplifiers for M-Sequence Radars Applications
We sincerely thank the Editor and Reviewers for their valuable comments and suggestions to enhance the quality of our manuscript. We have addressed all the comments and suggestions given by the reviewers and incorporated the same in the revised
REVIEWER #3
This work presents 6 UWB differencial amplifiers for applications in radars.
The 6 designs were fabricated and tested.
Nonetheless, there are few flaws in the manuscript that must be fixed before recommend the work for publication on these transactions.
The core architecture is the Cherry-Hooper structure, which is resonably described.
Comment 1.: On line 88, what is meant with "this chapter"?
Response 1.: Thank you for this comment, this is our mistake.
Comment 2.: The variables on figure 2 must be better associated to their equivalent counterparts in figure 1 (Vid/2 versus Vi1 and Vi2).
Response 2.: Thank you very much for your comment. We updated the figures 1 and 2, and we also checked others.
Comment 3.: The kappa parameter in line 127 must be clarified because the documentation in PDK does not provide a closed nunmber. Therefore, since this value was taken from a textbook to get the first kick towards the theoretical design and trim the W/L dimensions with simulations, thus, this must be clarified/described in the main body of manuscript.
Response 3.: Thank you for your comment, in text below we provide update:
The value of k' for 0.35 μm SiGe BiCMOS technologies is approximately 140 μA/V up to 200 μA/V [25], [26]. Rows(124-125)
Comment 4.: The authors done simulations for several Vdd and Vcm to access the reliability of the current mirrors and differencial amplifiers?+3
Response 4.: Thank you for your comment. We only simulate the current source together with the differential amplifiers, and watched their behavior together with amplifier.
Comment 5.: In line 143 it is stated the room temperature of 25 °C. Does the authors done simulations for other temperatures?
Response 5.:Thank you very much for your suggestion to improve the discussion. We only simulate the designs at 27°C. The process described in this document is qualified in the temperature range -40°C <= Tj <=125°C. SPICE models are valid in the temperature range -40°C< Tj <125°C (Tj specified as junction temperature). All the other measurements are done at T0 = 27°C. Rows (325)
Comment 6.: In equation (12) and line 188 it is stated a high input resistance. How high it is?
Response 6.: Thank you very much for your comments. Based on the Smith diagram and calculation of the input impedance of the emitter follower approximately dependent on output impedance multiplied by β input impedance of the emitter follower reaches units of kΩ, but at higher frequencies, it goes down due to parasitic capacities at the transistor model.
Comment 7.: The experimental details and photos of setups that allowed to obtain the 6 sets of results in table 1 must be provided and explained.
Response 7.: Thank you for your comments. We added informations about setups:
The block diagram and photo of probe-station measurement setup is shown in Figure 30. For S-parameter probe-station measurement Keysight N5183B vector analyzer with maximum frequency to 13.5 GHz the was used. There were used types of micro-probes, namely CASCADE MICTROTECH ACP40-AW-SG-100 with 40 GHz bandwidth and MPI TITAN T26V-SG0100 probe with 26 GHz bandwidth. Because the probe are only in SG (Signal-Ground) configuration, the quasi differential connection was applied with PE8212 inner outer DC-block capacitor, so that the probe can be connected to the differential input with 50 Ω. At the output only standard DC-block capacitor was used. The second output of the amplifier was unconnected and unmatched due to space limitation and no possibility to connect another probe. Such a probe-station best describes the actual parameters of the amplifiers as they are not affected by wire-bonding, packages and PCBs. The compression point and output voltage swing was measured on wire-bonded and assembled amplifier on PCB. The block diagram and photo of PCB measurement setup is shown in Figure 30. For compression measurement the signal generator Keysight N5183B and Agilent N9020A spectrum analyzer up to 26.5 GHz was used and in case of voltage swing measurement the spectrum analyzer was replaced by wideband oscilloscope Agilent DSO9404A with 20 GHz sampling. For DC-block on board wideband 520L103KT16T 100nF ceramic capacitors was used. In this case the second output was terminate with 50 Ω load, the second input was terminated directly to ground via wideband ceramic capacitor. The resulting comparison of all the parameters of the proposed amplifiers is presented in Table1. Rows (531-551)

Reviewer 4 Report
Comments and Suggestions for Authors
The paper presents different structures of differential amplifier for Radars applications. The modified Cherry-Hooper amplifier structures have been thoroughly analyzed. However, the following comments should be undertaken to improve the quality of the paper:
1) The paper demands careful proof-reading.
2) Equation 4 is obtained by multiplying equation (2) and equation (3).
3) Which technique has been used to design input and output matching networks. The information on CAD tool is missing.
4) Similar research works reported in literature should have been considered for comparison with modified Cherry-Hooper amplifier structures reported in the paper.
5) The efficiency performance of differential amplifier structures should also be reported.

Paper must be checked for English grammar.
Author Response
RESPONSE TO REVIEWER COMMENTS
Manuscript Number : sensors- 2878182
Article Type : Full Length Article
Article Title : Design and Realization of Ultra-wideband Differentials
Amplifiers for M-Sequence Radars Applications
We sincerely thank the Editor and Reviewers for their valuable comments and suggestions to enhance the quality of our manuscript. We have addressed all the comments and suggestions given by the reviewers and incorporated the same in the revised manuscript.
REVIEWER #4
The paper presents different structures of differential amplifier for Radars applications. The modified Cherry-Hooper amplifier structures have been thoroughly analyzed. However, the following comments should be undertaken to improve the quality of the paper:
Comment 1.: In view of the following instances of typos and grammar issues, the paper demands careful proof reading
Response 1.: Thank you for this comment. We have corrected typos and grammar issues.
Comment 2.: Equation 4 is obtained by multiplying equation (2) and equation (3)..
Response 2.: Thank you for this comment. We have corrected our statement that it is a multiplication. Row (100)
Comment 3.: Which technique has been used to design input and output matching networks. The information on CAD tool is missing.
Response 3.: Thank you very much for this comment. As mentioned, all circuits were designed and simulated in the CADENCE virtuoso development environment. The input and output matching was simulated with SP analysis in CADENCE Virtuoso design environment with ideal 50 Ω connected to amplifier, and these simulations were performed using RC-extract and approximate wire-bond induction. The input and output matching resistors were tuned using the smith diagram and the S11 linear graph. Rows (321-326 and 503)
Comment 4.: Similar research works reported in literature should have been considered for comparison with modified Cherry-Hooper amplifier structures reported in the paper.
Response 4.: Thank you for this comment for comparison we added one column to Table 1. With parameters of Cherry-hopper amplifier from [10]. See Table 1.
Comment 5.: The efficiency performance of differential amplifier structures should also be reported.
Response 5.: Thank you for this comment. We add Power Added Efficiency (PAE) at -10dBm input signal, the ratio of added RF power (RF output power minus RF input power) to DC power. See Table1.

Round 2
Reviewer 4 Report
Comments and Suggestions for Authors
The authors have adequately addressed my comments. As a result of it, paper has been improved.